# In Vitro Anti-Inflammatory and Antioxidant Activities of pH-Responsive Resveratrol-Urocanic Acid Nano-Assemblies

**DOI:** 10.3390/ijms24043843

**Published:** 2023-02-14

**Authors:** Heegyeong Song, Seok Kang, Ying Yu, Sung Yun Jung, Kyeongsoon Park, Sang-Min Kim, HaK-Jun Kim, Jae Gyoon Kim, Sung Eun Kim

**Affiliations:** 1Department of Systems Biotechnology, Chung-Ang University, Anseong 17546, Republic of Korea; 2Department of Physical Medicine and Rehabilitation and Nano-Based Disease Control Institute, Korea University Guro Hospital, #148, Gurodong-ro, Guro-gu, Seoul 08308, Republic of Korea; 3Department of Orthopedic Surgery and Nano-Based Disease Control Institute, Korea University Guro Hospital, #148, Gurodong-ro, Guro-gu, Seoul 08308, Republic of Korea; 4Department of Orthopedic Surgery, Korea University College of Medicine, Korea University Ansan Hospital, 123, Jeokgeum-ro, Danwon-gu, Ansan-si 15355, Republic of Korea

**Keywords:** resveratrol, urocanic acid, pH responsive, anti-inflammatory, antioxidant, nanoparticles

## Abstract

Inflammatory environments provide vital biochemical stimuli (i.e., oxidative stress, pH, and enzymes) for triggered drug delivery in a controlled manner. Inflammation alters the local pH within the affected tissues. As a result, pH-sensitive nanomaterials can be used to effectively target drugs to the site of inflammation. Herein, we designed pH-sensitive nanoparticles in which resveratrol (an anti-inflammatory and antioxidant compound (RES)) and urocanic acid (UA) were complexed with a pH-sensitive moiety using an emulsion method. These RES-UA NPs were characterized by transmission electron microscopy, dynamic light scattering, zeta potential, and FT-IR spectroscopy. The anti-inflammatory and antioxidant activities of the RES-UA NPs were assessed in RAW 264.7 macrophages. The NPs were circular in shape and ranged in size from 106 to 180 nm. The RES-UA NPs suppressed the mRNA expression of the pro-inflammatory molecules inducible nitric oxide synthase (iNOS), cyclooxygenase-2 (COX-2), interleukin-1β (IL-1β), and tumor necrosis factor-α (TNF-α) in lipopolysaccharide (LPS)-stimulated RAW 264.7 macrophages in a concentration-dependent manner. Incubation of LPS-stimulated macrophages with RES-UA NPs reduced the generation of reactive oxygen species (ROS) in a concentration-dependent manner. These results suggest that pH-responsive RES-UA NPs can be used to decrease ROS generation and inflammation.

## 1. Introduction

Osteoarthritis (OA), the most common degenerative joint disease, is characterized by degradation of the extracellular matrix (ECM), synovial inflammation, subchondral bone sclerosis, and regression of articular cartilage, and is a major cause of disability among the elderly worldwide [1,2,3]. The global incidence of OA is increasing due to the increase in obesity rates and resulting joint trauma [4,5]. The general consensus is that the key to treating this disease is to prevent the patients’ deterioration to severe stages that require surgery by suppressing the progression of inflammation and destructive processes in the early stages. However, most current treatments provide only pain relief and do not inhibit inflammatory activity or reverse cartilage damage [6,7]. Clinical management options for OA can be divided into non-pharmacological, pharmacological, and surgical measures [8,9]. Initial treatment typically comprises non-pharmacological strategies, including self-care, exercise, weight loss, aquatic exercise, and aerobic and resistance exercise [10,11]. The most commonly used drugs used for local, oral, and/or intra-articular pharmacological treatment are non-steroidal anti-inflammatory drugs (NSAIDs), COX-2 selective inhibitors, acetaminophen, paracetamol, corticosteroids, and hyaluronic acid (HA) [8,12,13], each of which has significant limitations. Generally, NSAIDs and COX-2 inhibitors are associated with gastrointestinal toxicity, cardiovascular side effects, nephrotoxicity, and cardiac nephrotoxicity [14]; long-term use of these substances may produce adverse effects on the digestive system and blood coagulation, which can alter immune function and worsen arthritis [15,16]. The materials used in intra-articular treatments are rapidly cleared and/or degraded and need to be administered frequently [17]. Long-term treatment with corticosteroids is associated with serious side effects, such as osteoporosis, osteonecrosis, hyperglycemia, and hypertension [18]. Therefore, alternative management strategies based on natural materials that have fewer side effects and can effectively inhibit cartilage degeneration are urgently required.

Resveratrol (RES), first extracted from the roots of *Veratrum grandiflorum* in 1940, is a polyphenolic compound found in many fruits and vegetables such as peanut bean sprouts, grapes, and peanuts [19,20]. The beneficial effects of RES include anti-inflammatory, free radical scavenger, immunomodulatory, antibacterial, and anti-diabetic activities, together with the inhibition of platelet aggregation and coagulation, and reduction in the risk of cardiovascular disease (CVD) [21,22,23,24]. The anti-inflammatory properties of RES are linked to the inhibition of TNF-α, IL-1β, and nitric oxide (NO) synthesis [25]. Previous studies have reported that RES also downregulates COX-2, IL-1β, and IL-6, which play important roles in OA by blocking nuclear factor-κB (NF-κB) signaling [26,27]. Other studies have confirmed that intra-articular injection of RES in rats with OA induced by monosodium iodoacetate (MIA) injection or in rabbits with arthritis induced by lipopolysaccharide (LPS) injection not only inhibited the expression of inflammatory factors but also prevented the degeneration of glycosaminoglycans (GAGs) [28,29].

Joint degeneration in OA is mainly due to inflammation, although the degree of inflammation is lower than that seen in rheumatoid arthritis (RA). The concentrations of COX-2, IL-1β, and TNF-α are significantly elevated in the synovial tissues of OA patients and contribute to OA progression [30]. In the context of OA, nanodrug delivery systems are being widely explored as options to provide the selective, controlled, and sustained delivery of drugs to inflammatory sites after intra-articular injection. Nanoparticles (NPs) injected into the joint space respond to physiological factors such as pH, reactive oxygen species (ROS), and enzymes. Although the pH range of the synovial membrane is 7.4 to 7.8 under normal physiochemical conditions, the pH in joints affected by OA may be as low as 6.0 due to the deposition of inflammatory metabolites in and around the joint tissues [31,32].

Given the acidic conditions within inflammatory sites, pH-responsive nanomaterials have been developed using fluorescent, porous nanofibers with high pH sensitivity that act as pH sensors and metal-organic frameworks to perform controlled drug delivery [33,34]. As an intermediate metabolite of histidine, urocanic acid (UA) exhibits a hydrophilic–hydrophobic transition at pH values around its pKa (~6.0) due to the presence of pH-sensitive imidazole groups that can be used as drug delivery carriers: at neutral pH, UA is amphiphilic and can self-assemble into NPs at concentrations greater than the critical aggregation concentration (CAC) due to hydrophobic interactions between urocanyl groups [35]; however, under acidic pH conditions (pH ~6.5), the imidazole groups are protonated (positively charged) and the nanostructures disassemble due to charge repulsion, allowing loaded drugs to be released [35,36,37].

In order to deliver drug molecules under low pH conditions and to effectively control inflammation in inflamed tissues, we designed pH-responsive RES-UA NPs that self-assemble due to interactions between hydrophobic RES and amphiphilic UA in the presence of surfactant (Figure 1A). The prepared RES-UA NPs were characterized to evaluate particle sizes and pH-responsive drug release. We also investigated the in vitro cytotoxicity and the anti-inflammatory and antioxidant properties of the RES-UA NPs. 

## 2. Results and Discussion

### 2.1. Characterization

Urocanic acid (UA), which has both imidazole and carboxyl groups, has been used as a pH-responsive molecule to adjust polymers by chemical modification [35,36,37]. Table 1 shows the particle size and zeta potential of RES-UA NPs under different pH conditions. After incubation for 24 h at pH 6.0, the hydrodynamic radius of RES-UA NPs was slightly elevated at 159.30 ± 36.80 nm (polydispersity index; PDI: 0.531 ± 0.05), compared to RES-UA NPs at pH 7.4 that had a hydrodynamic radius of 106.50 ± 31.00 nm (PDI: 0.545 ± 0.08). The hydrodynamic diameter of RES-UA NPs increased significantly to 188.70 ± 59.20 nm (PDI: 0.487 ± 0.08) at a lower pH (pH 5.0). As shown in Figure 1B, the increased size of RES-UA NPs at a lower pH was due to charge repulsion after protonation of the imidazole groups of UA in RES-UA NPs [36,37,38]. Furthermore, the assessed zeta potential of RES-UA NPs at pH 6.0 (−2.63 ± 0.25 mV) was lower than at pH 7.4 (−1.17 ± 0.31 mV). Under acidic pH conditions, RES-UA NPs released more RES and the carboxyl groups of UA molecules in RES-UA NPs were relatively more exposed than at pH 7.4. Thus, the determined zeta potential of RES-UA NPs (−2.80 ± 0.26 mV) at pH 5.0 was the lowest observed among the pH conditions tested. 

To assess the morphology of RES-UA NPs prepared at different pHs, we used transmission electron microscopy (TEM; Figure 1). The size of RES-UA NPs increased as the pH decreased, and their particle sizes measured by TEM were consistent and not significantly different from the results obtained using dynamic light scattering (DLS) because RES-UA NPs without hydrophilic shells did not shrink when dried. 

Figure 2 shows the FT-IR spectra of RES, UA, and RES-UA NPs. In the Fourier transform infrared (FT-IR) spectra, UA exhibited absorption peaks at 1660 cm^−1^ (C=O) and 1590 cm^−1^ (C=C) (Figure 2A). Characteristic absorption peaks at 3160 cm^−1^ and 3085 cm^−1^ corresponding to =C-H in UA were also present. With respect to RES, three peaks at 1610 cm^−1^, 1585 cm^−1^, and 1510 cm^−1^ corresponding to C=C bending, as well as two peaks at 3320 cm^−1^ corresponding to O-H and at 3015 cm^−1^ corresponding to =C-H were evident (Figure 2B) [39,40,41]. The FT-IR spectrum of RES-UA NPs exhibited similar peaks, including O-H, =C-H, C=O, and C=C peaks that are characteristic of UA and RES, indicating that RES-UA NPs consisted of UA and RES (Figure 2C).

### 2.2. Resveratrol Release from RES-UA NPs

We assessed drug-loading amounts and efficiency. Based on the standard curve for RES (Y = 0.036x + 0.0387, R^2^ = 0.9989), RES-UA NPs (1 mg) contained 71.05 ± 0.09 μg RES, and the RES loading efficiency was 78.10 ± 0.09%. The pH-responsive release of RES from RES-UA NPs was assessed at pH 7.4, 6.0, and 5.0. The results are shown in Figure 3. After 24 h, 43.10 ± 7.43% of RES was released from RES-UA NPs at pH 7.4, and 61.80 ± 6.62% and 73.26 ± 13.58% of RES was released at pH 6.0 and 5.0, respectively. The greater release of RES at pH 6.0 and pH 5.0 than at pH 7.4 was likely due to protonation and charge repulsion of the imidazole groups in UA under acidic conditions, leading to RES release from the NPs following disassembly of the NPs [35,36,37].

### 2.3. Cell Viability

To assess the cytotoxicity of RES-UA NPs, we determined cell viability using the CCK-8 assay. After 24 h treatment of RES-UA NPs at various concentrations, macrophage viability was maintained at over 85% for all tested concentrations (Figure 4a), indicating that RES-UA NPs are not cytotoxic to macrophages in cell culture. 

To create an inflammatory environment, we treated RAW 264.7 macrophage cells with LPS. Lipopolysaccharides are molecules located on the outer membranes of Gram-negative bacteria [25,26]. The presence of LPS stimulates host inflammatory responses that promote the production of chemokines, which stimulate macrophages [42]. LPS-stimulated macrophages produce high levels of pro-inflammatory mediators, resulting in enhanced inflammatory responses. 

RAW 264.7 cells were incubated with LPS for 24 h in the absence or presence of various concentrations of RES-UA NPs to assess the effects of RES-UA NPs in an inflammatory context. The survival of cells incubated with LPS was significantly lower than that of control cells, probably because inflammatory stimuli can interfere with cell proliferation (Figure 4b) [43]. However, the incubation of cells with (100 μg/mL) RES-UA NPs protected the cells against LPS-induced cell death. This result was consistent with the findings reported in a previous study in which various concentrations of RES (1, 5, and 10 μM) were found to protect macrophages against LPS-induced inflammation [24].

### 2.4. Effect of RES-UA NPs on Nitric Oxide Production in LPS-Activated RAW 264.7 Macrophages

Nitric oxide (NO) production is associated with the uptake of L-arginine, which is a substrate for NO synthase in LPS-treated macrophages [24]. We assessed NO production in LPS-activated RAW 264.7 cells in the presence or absence of RES-UA NPs using Griess reagent. As shown in Figure 5, untreated (control) cells secreted very little nitric oxide, whereas LPS-activated macrophages displayed elevated NO production. In LPS-stimulated cells incubated with RES-UA NPs (100 μg/mL), NO production was significantly lower than in LPS-stimulated cells (* *p* < 0.05). RES suppressed the LPS-induced formation of NO in macrophages in a concentration-dependent manner [24]. Consistent with previous studies, RES-UA NPs effectively reduced NO production in LPS-stimulated cells [24,44,45].

### 2.5. Effect of RES-UA NPs on Inflammatory Gene Expression in LPS-Treated RAW 264.7 Cells

The production of NO in macrophages is controlled by inducible NOS (iNOS) [46,47]. In LPS-induced macrophages, the production of various pro-inflammatory mediators and cytokines, including COX-2, IL-1β, IL-6, TNF-α, and prostaglandin (PGE2), is elevated [24,46]. As shown in Figure 6, LPS-stimulated cells showed markedly elevated expression of mRNA for *COX-2*, *IL-1β*, *iNOS*, and *TNF-α* compared with control group cells. In comparison, *COX-2*, *IL-1β*, *iNOS*, and *TNF-α* gene expression in cells incubated with RES-UA NPs was reduced in a concentration-dependent manner. The expression of *COX-2, IL-1β,* and *TNF-α* in cells incubated with RES-UA NPs (50 μg/mL) was lower than in LPS-activated cells. Furthermore, *COX-2, IL-1β, iNOS,* and *TNF-α* mRNA expression in cells incubated with RES-UA NPs (100 μg/mL) was downregulated compared to cells incubated with RES-UA (10 μg/mL) and RES-UA (50 μg/mL). RES can inhibit the formation of pro-inflammatory cytokines such as COX-2, IL-1β, iNOS, and TNF-α, in a dose-dependent manner [24,48]. Consistent with previous studies [24,45,48], we demonstrated that RES-UA NPs had a noticeable anti-inflammatory effect by inhibiting the mRNA expression of pro-inflammatory cytokines.

### 2.6. Antioxidant Effects of RES-UA NPs and ROS Scavenging Assay at the Cellular Level

The antioxidant activity of RES-UA NPs was determined using the ABTS assay [38]. As shown in Figure 7a, RES-UA NPs had ROS scavenging activities at all tested concentrations. Compared to the ROS scavenging activity of Vitamin C (50 μg/mL) at pH 7.4, RES-UA NPs showed lower ROS scavenging activities at all tested concentrations. However, RES-UA NPs exhibited increased ROS scavenging activity with decreasing pH values because RES-UA NPs could release higher amounts of RES from the NPs, as shown in Figure 3. Additionally, RES-UA NPs displayed dose-dependent ROS scavenging activity at pH 5.0, which was associated with the amount of RES released from the NPs, consistent with the findings of previous studies [49,50]. 

Reactive oxygen species (ROS) are highly reactive and unstable molecules that contain oxygen produced during normal and disease-associated metabolic processes; the superoxide anion (·O_2_^−^), hydrogen peroxide (H_2_O_2_), hydroxyl radicals, and singlet oxygen are all ROS [51,52,53,54]. LPS stimulates ROS production and inflammatory cytokine release within hours [52,54]. The secreted inflammatory cytokines accelerate intracellular ROS accumulation by encouraging the inflow of macrophages into tissues [47]. In addition, the overproduction of ROS causes oxidative stress, which eventually leads to cell death.

Macrophages were pre-incubated with various concentrations of RES-UA NPs for 1 h, followed by the addition of LPS (100 ng/mL) to induce oxidative stress and a 24 h incubation to assess the antioxidant activity of RES-UA NPs at the cellular level. At the end of the incubation, intracellular ROS in cells was monitored using the fluorescent dye DCF-DA and a confocal laser scanning microscope (CLSM; Carl Zeiss, Germany). As shown in Figure 7b, control cells (no RES-UA NPs or LPS added) did not fluoresce, whereas LPS-stimulated cells showed strong fluorescence. The addition of various concentrations of RES-UA NPs (10, 50, or 100 μg/mL) resulted in a remarkable concentration-dependent reduction in fluorescence intensity. To further quantify ROS generation, we measured DCFDA fluorescence intensity. As shown in Figure 7c, LPS-stimulated cells showed higher fluorescence intensity than control cells, while cells incubated with REA-UA NPs showed a concentration-dependent decrease in fluorescence intensity. In other words, the fluorescence intensity of cells incubated with 50 or 100 μg/m RES-UA NPs was much lower than that of LPS-stimulated cells (*** *p* < 0.001). These findings suggest that the antioxidant effects of RES-UA NPs prevented ROS generation in macrophages in a concentration-dependent manner, consistent with the results of previous studies [38,55].

## 3. Materials and Methods

### 3.1. Preparation of Resveratrol and Urocanic Acid Nanoparticles

To prepare the resveratrol (RES; Carbosynth Ltd., Staad, Switzerland) and urocanic acid (UA; Sigma-Aldrich, St. Louis, MO, USA) nanoparticles, we used a single-step emulsion solvent evaporation method and ultrasonication. First, UA (200 mg) was dissolved in 10-mL vials containing 5 mL N,N-dimethylformamide (DMF; Sigma-Aldrich), followed by addition of RES (20 mg) and vigorous mixing for 1 min. Afterward, the mixture was added to 10% polyvinyl alcohol (PVA, Mw = 13,000–23,000, 98% hydrolyzed; Sigma-Aldrich) solution (3 mg PVA in 30 mL deionized water) at a flow rate of 0.1 mL/min using an NE-100 syringe pump (New Era Pump Systems Inc., Farmingdale, NY, USA) while stirring at 500 rpm for 3 min. After incubation for 3 min, the resultant mixture was re-dispersed using a probe ultrasonicator (55 W; Korea Process Technology Co., Ltd., Seoul, Republic of Korea) with a pulse function (pulse on/off = 10/2 s) in an ice bath for 30 min and subsequently incubated overnight at 25 °C. After incubation, the RES-UA NP solutions were rinsed three times with distilled water (DW) by centrifuging at 3000 rpm for 10 min. Finally, the RES-UA NP solutions were freeze-dried using an FD8515 freeze dryer (IlShinBioBase Co., Ltd., Dongducheon-si, Gyeonggido, Republic of Korea).

### 3.2. Characterization

To assess changes in morphology according to pH, we incubated RES-UA NPs (1 mg/mL) with 5 mL of phosphate-buffered saline (PBS; Welgene Inc., Gyeongsan-si, Gyeongsangbuk-do, Republic of Korea) under different pH conditions (pH 7.4, 6.0, or 5.0) at 37 °C for 24 h. The hydrodynamic size and zeta potential of the RES-UA NPs were measured under the above conditions. Samples were further characterized by field emission transmission electron microscopy (FE-TEM; JEM-F200; JEOL Ltd., Tokyo, Japan). The RES-UA NPs were sonicated in ethanol (C_2_H_6_OH; DaeJung Chemical & Metals, Siheung-si, Gyeonggi-do, Republic of Korea) by probe ultrasonication for 1 h, followed by deposition onto carbon-coated TEM grids and drying in a drying oven to remove residual ethanol. The morphology of the NPs was assessed by FE-TEM at a running voltage of 200 kV. The hydrodynamic size and zeta potential of the RES-UA NPs were determined at 24 h using a nanoparticle analyzer (SZ-100V2; HORIBA, Kyoto, Japan) at a 90° angle based on dynamic light scattering (DLS) fundamentals. Regarding the calculations, each group (100 μg) was dispersed in 10 mL DW by probe sonication for 10 min over ice. Next, each scattered group was transferred to a disposable container and measured as described above. Resveratrol, urocanic acid, and RES-UA NPs were characterized using Fourier transform infrared (FT-IR) spectroscopy (Shimadzu 8400S; Kyoto, Japan). The FT-IR spectra were acquired using KBr powder at a resolution of 4 cm^−1^ and a wavelength range from 4000 to 400 cm^−1^. To quantify the drug-loading amount and efficiency, we dissolved the RES-UA NPs (1 mg) into 1 mL of methanol for at least 1 h, and the absorbance was measured at 307 nm using a multimode microplate reader (Varioskan™; Thermo Scientific, Walthan, MA, USA). Then, the drug amount and efficiency were determined based on the standard curve for RES (Y = 0.0306*x* + 0.0387, R^2^ = 0.9989; RES concentration range of standard curve: 0 to 20 μg/mL). 

### 3.3. In Vitro pH-Responsive RES Release Behavior of RES-UA NPs

Freeze-dried RES-UA NPs were investigated by dialysis at different pH values (pH 7.4, 6.0, or 5.0) to assess the pH-responsive release profiles of RES-UA NPs. The RES-UA NP (2 mL at 1 mg/mL) solution was placed in a dialysis tube (cellulose membrane; MWCO of 3500 Da; Spectrum Laboratories, CA, USA), followed by immersion in a solution containing 30 mL of PBS buffer at a pH of 7.4, 6.0, or 5.0 and then incubation at 37 °C with continuous shaking at 100 rpm. At various time intervals (1, 3, 6, 12, 24, 48, and 72 h), 4 mL of the release medium was removed and replaced with 4 mL of the same amount of fresh release medium. The amount of released RES was monitored using a multimode microplate reader at a wavelength of 307 nm. The amount of RES released was calculated based on the standard curve for RES (Y = 0.0306x + 0.0387, R^2^ = 0.9989; RES concentration range of standard curve: 0 to 20 μg/mL).

### 3.4. Cell Viability Assessment

To assess the cytotoxicity of RES-UA NPs, we plated RAW 264.7 macrophages (Korea Cell Line Bank, Seoul, Republic of Korea) at a density of 3 × 10^4^ cells-per-well in 96-well culture plates (Welgene Inc.) and cultivated them in DMEM (Thermo Fisher Scientific, Rockford, IL, USA) containing 10% FBS (Thermo Fisher Scientific) and 1% antibiotics (Thermo Fisher Scientific) in an incubator at 37 °C with 5% CO_2_ for 24 h. Then, the cells were incubated with RES-UA NPs (0, 10, 50, or 100 μg/mL) for 24 h, and CCK-8 (Cell Counting Kit-8, Dojindo, Inc., Rockville, MD, USA) reagent for 1 h in the dark at 37 °C, and well supernatants were transferred into 96-well plates. 

To determine if the RES-UA NPs were able to protect against lipopolysaccharide-induced (LPS; Sigma-Aldrich) cytotoxicity, cells were plated at a density of 3 × 10^4^ cells-per-well in a 96-well plate and incubated for 24 h. They were then pre-incubated with RES-UA NPs (0, 10, 50, and 100 μg/mL) for 1 h, followed by stimulation with LPS (100 ng/mL) for 24 h. Then, cells were rinsed three times with PBS, followed by the addition of CCK-8 reagent and incubation for 1 h in the dark. At the end of this incubation period, supernatants were transferred into the wells of a new 96-well plate.

The absorbance of cells after the addition of RES-UA NPs with or without subsequent LPS stimulation was recorded using a multimode microplate reader at a wavelength of 450 nm. Untreated cells cultured under the same conditions as the treated cells were used as the control group. Four samples per group were analyzed, and all experiments for each time period were repeated three times.

### 3.5. Anti-Inflammatory Potential

#### 3.5.1. Investigation of Nitric Oxide Production

The RAW 264.7 cells (2 × 10^5^ per well) were plated into 24-well plates and incubated for 24 h. Cells were rinsed twice with PBS, followed by the addition of serum-free medium containing LPS (100 ng/mL) and different concentrations of RES-UA NPs (0, 10, 50, or 100 μg/mL) and incubation for another 24 h at 37 °C. The supernatants were harvested to analyze nitrite accumulation using Griess reagent. In brief, supernatant (200 μL) was added to an Eppendorf tube, followed by the addition of an equal volume of Griess reagent (1% (*w*/*v*) sulfanilamide (DaeJung Chemical & Metals) in 5% (*v*/*v*) phosphoric acid and 0.1% (*w*/*v*) naphthylethylenediamine-HCl) and incubation for 10 min in the dark. At the end of this incubation, optical density at 540 nm was determined using a multimode microplate reader. The supernatant harvested from untreated cells was used as the blank in all experiments. The amount of nitrite in the samples was determined by referring to a sodium nitrite standard curve (Y = 0.0460*x* + 0.0204, R^2^ = 0.996). 

#### 3.5.2. Real-Time Reverse Transcription-Polymerase Chain Reaction Determination of Transcript Levels of Target Genes

The in vitro anti-inflammatory effects of RES-UA NPs in LPS-activated macrophages were investigated by quantifying the mRNA expression of pro-inflammatory mediators using real-time PCR. Cells were plated at a density of 2 × 10^5^ cells-per-well in 24-well cell culture plates and incubated for 24 h. The cells were then stimulated with LPS (100 ng/mL) in the presence or absence of different concentrations of RES-UA NPs (0, 10, 50, or 100 μg/mL) for 24 h. Thereafter, cells were rinsed with PBS three times, followed by RNA extraction using an RNeasy Plus mini kit (Qiagen, Valencia, CA, USA). cDNA was obtained from total RNA (1 μg) using AccuPower RT PreMix (Bioneer, Daejeon, Republic of Korea) after the measurement of total RNA. The sequences of the pro-inflammatory mediator primers were as follows: COX-2: (F) 5′-GCG ACA TAC TCA AGC AGG AGC A-3′, (R) 5′-AGT GGT AAC CGC TCA GGT GTT G-3′; IL-1β: (F) 5′-TGG ACC TTC CAG GAT GAG GAC A-3′, (R) 5′-GTT CAT CTC GGA GCC TGT AGT G-3′; iNOS: (F) 5′-GAC TTT CCA AGA CAC ACT TCA C-3′, (R) 5′-TTC GAT AGC TTG AGG TAG AAG C-3′; TNF-α: (F) 5′-AAG CCT GTA GCC CAC GTC GTA-3′; (R) 5′-GGC ACC ACT AGT TGG TTG TCT TTG-3′. An ABI7300 Thermal Cycler was used to perform thermocycling, and SYBR Green PCR master mix was used for real-time monitoring of transcript levels (Applied Biosystems, Foster City, CA, USA). Glyceraldehyde 3-phosphate dehydrogenase (GAPDH) was used to normalize the expression of pro-inflammatory mediator genes.

### 3.6. Antioxidant Effects of RES-UA NPs

#### 3.6.1. In Vitro Antioxidant Capacity

To evaluate the antioxidant capacities of various concentrations of RES-UA NPs in PBS with different pH conditions (pH 7.4, 6.0, and 5.0), we performed the ABTS (2,2′-azino-bis(3-ethylbenzothiazoline-6-sulfonic acid)) assay, as described previously [38]. L-ascorbic acid (Vit-C; Tokyo Chemical Industry Co., Ltd., Tokyo, Japan) was used as a positive control. The ABTS radical solution was generated by mixing equal volumes of 14 mM ABTS solution and 4.9 mM K_2_S_2_O_8_ (potassium persulfate) solution for 14 h at 25 °C in the dark. The resulting mixture was diluted with PBS solution until the absorbance was 0.7 ± 0.02 at 734 nm. Vit-C (50 μg/mL) was dissolved in PBS (pH 7.4) and then diluted in ABTS solution. Various concentrations of RES-UA NPs (10, 50, and 100 μg RES/mL) were prepared in PBS with a pH of 7.4, 6.0, or 5.0, and then the diluted ABTS solution, Vit-C solution, and sample solution were combined at a 3:1 *v*/*v* ratio and left to react in the dark for 2 h. The absorbance of the reaction solution was monitored from 300 to 900 nm using a multimode microplate reader. At 734 nm, the ABTS scavenging of the various concentrations of RES-UA NPs and Vit-C was calculated using the following equation: Scavenging activity = [(Abs_con_ − Abs_sam_)/Abs_con_] × 100, where Abs_con_ is the absorbance of the control, and Abs_sam_ is the absorbance of the sample in PBS with different pH values (pH 7.4, 6.0, and 5.0).

#### 3.6.2. ROS Scavenging Ability of RES-UA NPs

To assess whether RES-UA NPs could protect cells against oxidative stress, we investigated intracellular ROS generation using an ROS-responsive fluorescence indicator, 2,7-dichlorofluorescein diacetate (DCF-DA; Sigma-Aldrich). Macrophages were deposited at a density of 5 × 10^4^ on cover glasses, and these were then placed into the wells of 24-well plates followed by a 24-h incubation. Cells were washed three times with PBS and pre-exposed to RES-UA NPs (0, 10, 50, or 100 μg/mL) in serum-free medium for 1 h, followed by stimulation with LPS (100 ng/mL) for 24 h. Cells were stained with DCFDA (25 μM) in the dark for 30 min and then fixed with 3.7% paraformaldehyde (BioSesang Co., Seongnam-si, Gyeonggi-do, Republic of Korea) for 30 min. After fixation, 4′,6-diamidino-2-2phenylidole (DAPI; Tokyo Chemical Industry Co., Ltd., Tokyo, Japan) and Alexa^®^ Fluor 594-labeled wheat-germ agglutinin (Thermo Fisher Scientific Inc.) were used to stain cell nuclei and membranes, respectively. Each sample was examined under a confocal laser scanning microscope (CLSM; Carl Zeiss, Oberkochen, Germany). Fluorescence intensity was measured using a multimode plate reader at excitation and emission wavelengths of 485 and 535 nm, respectively, to further quantify ROS formation.

### 3.7. Statistical Analysis

All data are reported as means ± standard deviations. Statistical analyses were performed using a one-way ANOVA with the post hoc Holm–Sidak test in SigmaPlot 12.0 (Systat Software, Inc., San Jose, CA, USA). *p* values less than 0.01 were considered statistically significant.

## 4. Conclusions

In this study, pH-responsive self-assembled RES-UA nanoparticles were successfully fabricated using single-emulsion solvent evaporation and ultrasonication. The release pattern of RES-UA NPs was faster in slightly acidic environments than in neutral environments. The RES-UA NPs had anti-oxidative and anti-inflammatory effects in LPS-stimulated RAW 264.7 macrophages. The RES-UA NPs also significantly reduced the production of NO in LPS-activated cells in a concentration-dependent manner. In addition, RES-UA NPs remarkably diminished the mRNA expression of pro-inflammatory molecules (COX-2, IL-1β, iNOS, and TNF-α). Furthermore, RES-UA NPs effectively scavenged ROS in cells in a concentration-dependent fashion. Accordingly, RES-UA nanoparticles are promising materials for the treatment of inflammatory conditions such as osteoarthritis.

## Data Availability

Not applicable.

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
