# Peer review of "In Vitro Anti-Inflammatory and Antioxidant Activities of pH-Responsive Resveratrol-Urocanic Acid Nano-Assemblies"

_ijms, 2023, doi:10.3390/ijms24043843_

Round 1
Reviewer 1 Report
1. "nanoparticles"should be added as Keywords.
2. The font type and font size should be the same for the horizontal and vertical coordinates of all figures.
3. The numbers and units of the yellow ruler in Figure 7 are unclear.
4. The room temperature (RT) should be specified.
5. Is there any difference between the particle size measured by SEM and that measured by DLS? Why?
6. Is the Zeta potential of nanoparticles consistent with the results of other scholars? Is the value of Zeta potential reasonable?
7. Some references need to be updated.
Author Response
According to the reviewers' comments, we corrected the main text or some figures in the revised version. We attached the responses to reviewers' comments. Please the attached files.

Reviewer 2 Report
Authors have investigated a pH-responsive resveratrol-urocanic acid directed for nano-assemblies, their material exhibits anti-inflammatory and antioxidant activities properties. The output of the research is interesting and presented results are rigorous. However, various flaws minimize the quality of this research paper and need to be addressed. I recommend Major revisions of this manuscript and advise authors to review below points to enhance the quality of their manuscript before publication in International Journal of Molecular Sciences.
To address,
1) To expend your introduction, I suggest the following references (ACS Appl. Mater. Interfaces, 2017, 9, 16381-16396; 10.1021/acsami.7b00970) (J Nanobiotechnol, 2020, 18, 139; 10.1186/s12951-020-00694-3).
2) While the introduction is well-written, the transition toward the purpose of your study should be much smoother and explained, otherwise the introduction or the objective of your work would have little meaning for the readers.
3) The experimental process could be illustrated.
4) What could be the reason behind the pH-dependent size of the particle?
5) Do you have any references for the argumentation related to the FT-IR data
6) The Figure 3 choice of the itemization is not optimized.
7) Figure 4b, with and without LPS sample bar diagram could be distinguished to increase the visibility of the information.
8) Correct English here and there
Author Response

(The authors gave the same response as above.)

Reviewer 3 Report
The work presented by Heegyeong Song et al. is scientifically valid. The experimental design is well structured, and the scientific language is used appropriately. However, there are some issues must be checked in order to ameliorate the entire work and make it suitable for publication. First, the English must be improved both in terms of syntax and sentence structure. Then the results should be supported by a consistent discussion and references should be checked. Furthermore, here are some comments in detail.
Check acronyms (e.g. Resveratrol, Urocanic acid, ecc.), the manufacturers of some materials such as reagents are missing, please carefully check and add them.
Abstract: this section could be improved by adding the research background.
Lines 104-106: I strongly suggest to Authors adding discussion about the meaning of why Z Potential is related to the pH values.
Lines 121-134: As above, the Authors just reported the results without proper discussion. Please improve this part.
Lines 145-146: the quantification of Resveratrol is misleading, please improve this sentence.
Line 163: this sentence is redundant and not fluent, please improve it.
Line 196: this sentence is redundant and misleading.
Line 286: In which solvent was prepared the PVA solution? Please add it.
Lines 311-313: This part is misleading. The Authors just reported a simple centrifugation method to determine the encapsulation efficiency of Resveratrol but the description of this analysis should be improved.
Lines 321-323: The Resveratrol calibration curve is missing, please add it specifying the drug concentration range, curve equation and regression coeffiecient.
Lines 355-357: As above, the sodium nitrite standard curve is missing. Please add it.
Author Response

(The authors gave the same response as above.)

Reviewer 4 Report
The manuscript accounts of a potentially useful study demonstrating anti-inflammatory and antioxidant properties of resveratrol-urocainic acid pH-responsive nanoparticles in vitro. The presented results point to a possibility of a therapeutic application of such nanoparticles in the treatment of osteoarthritis.
Title: It could be indicated in the title that it is an vitro study; the present form of the title may be understood as promising also in vivo results.
I wonder about the relevance of the ABTS assay. The assay was done in PBS, pH 7.4 and apparently measured a fraction of the antioxidant activity of NPs, which could be greater after total RES release (e. g. at lower pH or in a hypotonic medium). Could this activity can be standardized in relation to a known antioxidant such as Trolox or ascorbic acid?
Lines 180-181: Please indicate the concentration of NPs which was protective (statistically significant effect)
Line 192: “through the degradation of L-arginine” is perhaps not the optimal term; L-Arg is a substrate for NO production but not especially degraded
Line 220: “compared to other groups”, all other groups?
Line 244: “(O2-.)”, the last symbol should be in superscript, not in a subscript
Figure 7A: please indicate statistical significance of differences
Line 286: Tough it is rather obvious, please explain the acronym
Author Response

(The authors gave the same response as above.)

Round 2
Reviewer 3 Report
The authors followed the recommendations of the reviewers and improved the overall quality of the work. However, English need to be improved yet.
Author Response
Thank you for the comment. As reviewer indicated, we edited English of our revised manuscript via the professional editing company, eWorldEditing, Inc.
